# Evaluation of Astatine-211-Labeled Fibroblast Activation Protein Inhibitor (FAPI): Comparison of Different Linkers with Polyethylene Glycol and Piperazine

**DOI:** 10.3390/ijms24108701

**Published:** 2023-05-12

**Authors:** Ayaka Aso, Hinako Nabetani, Yoshifumi Matsuura, Yuichiro Kadonaga, Yoshifumi Shirakami, Tadashi Watabe, Taku Yoshiya, Masayoshi Mochizuki, Kazuhiro Ooe, Atsuko Kawakami, Naoya Jinno, Atsushi Toyoshima, Hiromitsu Haba, Yang Wang, Jens Cardinale, Frederik Lars Giesel, Atsushi Shimoyama, Kazuko Kaneda-Nakashima, Koichi Fukase

**Affiliations:** 1Department of Chemistry, Graduate School of Science, Osaka University, 1-1 Machikaneyama, Toyonaka 560-0043, Osaka, Japan; ayaka0722lv@gmail.com (A.A.); nabetanih22@chem.sci.osaka-u.ac.jp (H.N.); matsuuray16@chem.sci.osaka-u.ac.jp (Y.M.); ashimo@chem.sci.osaka-u.ac.jp (A.S.); koichi@chem.sci.osaka-u.ac.jp (K.F.); 2Department of Nuclear Medicine and Tracer Kinetics, Osaka University Graduate School of Medicine, 2-2 Yamadaoka, Suita 565-0871, Osaka, Japan; kadonaga.yuuichirou.med@osaka-u.ac.jp; 3Division of Science, Institute for Radiation Sciences, Osaka University, 1-1 Machikaneyama, Toyonaka 560-0043, Osaka, Japan; yoshifumi_shirakami@irs.osaka-u.ac.jp (Y.S.); toyo@irs.osaka-u.ac.jp (A.T.); frederik.giesel@med.uni-duesseldorf.de (F.L.G.); 4Peptide Institute, Inc., 7-2-9 Saito-asagi, Ibaraki 567-0085, Osaka, Japan; t.yoshiya@peptide.co.jp (T.Y.); mochizuki@peptide.co.jp (M.M.); 5Institute for Protein Research, Osaka University, 3-2 Yamadaoka, Suita 565-0871, Osaka, Japan; 6Radioisotope Research Center, Institute for Radiation Sciences, Osaka University, 2-4 Yamadaoka, Suita 565-0871, Osaka, Japan; ooe@rirc.osaka-u.ac.jp; 7Research Center for Ultra-High Voltage Electron Microscopy, Osaka University, 7-1 Mihogaoka, Ibaraki 567-0047, Osaka, Japan; saito@uhvem.osaka-u.ac.jp; 8R&D Division, Alpha Fusion Inc., 10-1 Mihogaoka, Ibaraki 567-0047, Osaka, Japan; n-jinno@irs.osaka-u.ac.jp; 9RIKEN Nishina Center for Accelerator-Based Science, 2-1 Hirosawa, Wako 351-0198, Saitama, Japan; haba@riken.jp (H.H.); wangyang@impcas.ac.cn (Y.W.); 10Department of Nuclear Medicine, University Hospital Düsseldorf, Moorenstr. 5, 40225 Düsseldorf, Germany; jens.cardinale@med.uni-duesseldorf.de; 11Core for Medicine and Science Collaborative Research and Education, Forefront Research Center, Graduate School of Science, Osaka University, 1-1 Machikaneyama, Toyonaka 560-0043, Osaka, Japan

**Keywords:** FAP alpha, astatine-211, iodine-131, theranostics, PEG-linker, piperazine-linker, cancer

## Abstract

Fibroblast activation proteins (FAP) are overexpressed in the tumor stroma and have received attention as target molecules for radionuclide therapy. The FAP inhibitor (FAPI) is used as a probe to deliver nuclides to cancer tissues. In this study, we designed and synthesized four novel ^211^At-FAPI(s) possessing polyethylene glycol (PEG) linkers between the FAP-targeting and ^211^At-attaching moieties. ^211^At-FAPI(s) and piperazine (PIP) linker FAPI exhibited distinct FAP selectivity and uptake in FAPII-overexpressing HEK293 cells and the lung cancer cell line A549. The complexity of the PEG linker did not significantly affect selectivity. The efficiencies of both linkers were almost the same. Comparing the two nuclides, ^211^At was superior to ^131^I in tumor accumulation. In the mouse model, the antitumor effects of the PEG and PIP linkers were almost the same. Most of the currently synthesized FAPI(s) contain PIP linkers; however, in our study, we found that PEG linkers exhibit equivalent performance. If the PIP linker is inconvenient, a PEG linker is expected to be an alternative.

## 1. Introduction

The most important aspect of molecular-targeted cancer therapy is the selection of excellent molecular targets. Cancer specificity must be ensured in order to prevent adverse effects. Cancer tissue is a heterogeneous population in which cancer cells and various other cell types are mixed. Therefore, especially in cancer types that possess stromal cells that play a wall-like role, it is difficult to deliver drugs to them. By targeting fibroblast-activated protein (FAP), which is also a marker for stromal cells, we may be able to efficiently target cancer tissues. 

FAP inhibitor (FAPI), which has recently attracted attention as a cancer-targeting molecule, causes the accumulation of FAP, which is overexpressed in the tumor stroma and is involved in cancer growth. From a structural perspective, the presence of carbonitrile on the pyrrolidine ring of FAPI increases its affinity for FAP [1,2]. Furthermore, Jansen et al. argued that the quinoline moiety of FAPI forms a cation–π interaction with the guanidine moiety of Arg123 in FAP [3]. Lindner et al. compared the EC_50_ of FAPI(s) using competition experiments and found that the pharmacokinetics changed significantly according to the structure of the linker [4]. PET imaging using ^68^Ga-labeled FAPI-02, FAPI-04, and FAPI-13 revealed favorable tumor accumulation in vivo in HT-1080-FAP xenografted mice. Diagnostic positron emission tomography/computed tomography (PET/CT) scans using FAPI have been demonstrated to be useful for the development of clinical applications in patients [5,6]. Using ^68^Ga-labeled FAPI-02 and FAPI-04, excellent tumor accumulation was observed in a mouse sarcoma model and in pancreatic cancer patients [4,7]. In addition, new FAPI(s) (e.g., FAPI-21, FAPI-34, FAPI-46, and FAPI-74) have been developed to further improve their beneficial effects [8,9,10,11,12,13,14]. The ^68^Ga-Labeled FAPI dimer linked by polyethylene glycol (PEG) with DOTA exhibited outstanding tumor uptake and longer tumor retention than ^68^Ga-FAPI-46 [15]. ^18^F-labeled FAPI with a PEG linker without piperazine (PIP) has also been employed for clinical PET/CT imaging and nasopharyngeal cancers, in addition to ^18^F-FDG [16]. ^18^F-NOTA-FAPI has also been used for PET/CT imaging of patients [17]. Biodistribution of ^68^Ga-DOTA.SA.FAPI in patients was also investigated, and clear accumulation was shown by PET/CT imaging [18]. As it was possible to clearly visualize the cancerous lesion, it is speculated that [^68^Ga] Ga-DATA5m. SA.FAPI is a promising substrate for PET/CT imaging [19]. As the clinical application of FAPI-PET steadily progresses, an efficient and stable supply of ^18^F-labeled FAPI is urgently needed. Recently, we developed an automated radiosynthesis method for [^18^F]AIF-FAPI-74 [20]. 

FAPI, which was created by taking advantage of the cancer specificity of FAP, has been shown to have excellent performance as a PET/CT probe [6]. Thus, FAPI is a promising cancer-treating molecule. We synthesized ^177^Lu-labeled FAPI-46 (^177^Lu-FAPI) and administered it to PANC-1 xenograft mice [21]. ^177^Lu-FAPI showed a strong antitumor effect with 30 MBq of radioactivity. We also labeled FAPI-04 with ^225^Ac as an α-emitter and administered it to PANC-1 xenograft mice [22]. Although a low radioactivity level of ^225^Ac was used, excellent anti-tumor activity was achieved owing to the irradiation of particles, which have high linear energy transfer (LET). Furthermore, we focused on ^211^At as an emitter. ^211^At has a relatively short half-life of 7.2 h, and exhibits a simple decay chain without harmful daughter nuclides (^207^Bi and ^207^Pb). Therefore, if α nuclear medicine treatment becomes widely used, it is expected that the length of hospital stay would be shortened. Hospitalization may not be necessary in the future. If so, α nuclear medicine treatment will be able to further contribute to improving the quality of life of cancer patients. 

Our group has developed various ^211^At-labeled drug candidates such as [^211^At] NaAt, ^211^At-PA, ^211^At-PSMA, ^211^At-AAMT, and ^211^At-AuNP. The usefulness of ^211^At should also be demonstrated using FAPI. In 2022, Ma et al. first reported the antitumor activity of ^211^At-FAPI in U87MG xenograft mice [23]. They also claimed that no toxicity was observed in the kidneys, liver, stomach, or thyroid tissue. We also synthesized ^211^At-FAPI(s) using dihydroxyboryl astatine substitution reaction [24]. We have previously established an astatination method that does not require toxic reagents [24,25]. In this study, we investigated the usefulness of ^211^At-FAPI(s) synthesized using an established safe method and a linker in the FAPI structure.

## 2. Results

### 2.1. Establishment of FAPI1 to 5

PIP has been mainly employed as a linker for FAPI to improve its hydrophilicity. However, the PIP linker FAPI is degraded in the body [16]. The PEG linker is more stable than the PIP linker. Moreover, the production of byproducts during ^211^At treatment can be suppressed in the absence of an amino group. To confirm the effectiveness of the PEG linker FAPI, we synthesized four PEG linker FAPI(s). Their structures are illustrated in Figure 1a. FAPI1 has a simple PEG linker. In FAPI2, to reserve the space for the FAP binding site, the length of the PEG linker was extended to increase the distance between the radionuclide-labeling site and the biologically active site. FAPI3 increases stability by adding Fluorine, and FAPI4 decreases lipid solubility and increases urinary excretion by adding glucose. We also synthesized a PIP linker, FAPI, to compare its efficacy with that of the PEG linker, FAPI(s). The FAPI(s) were labeled with ^211^At or ^131^I at the same site in their structures (Figure 1b,c). The radiochemical purities (RCPs) of ^211^At-FAPI1, 2, 3, 4, and ^211^At-FAPI5 were 100%, 100%, 96%, 78%, and 95%, and those of ^131^I-FAPI1 and ^131^I -FAPI5 were 91% and 99%, respectively. The labeling efficiency of ^211^At-labeled FAPI with a PEG linker decreased with increasing complexity. The purity of ^131^I-labeled FAPI1 was lower than FAPI5.

The synthesis results and labeling procedures for each FAPI are shown in the Appendix A.

### 2.2. Comparison of Cellular Uptake between FAPI1 to 5

In an uptake experiment using HEK293 cells (Figure 2), the uptake of ^211^At-FAPI2 and 3 was slightly lower, whereas that of ^211^At-FAP1, 4, and 5 appeared to be the same. Because FAPI1–4 are PEG linker compounds and FAPI5 is a PIP linker compound, FAPI1, the simplest PEG linker, and FAPI5, a PIP linker, were compared. The difference was not notable in A549 cells (FAPα/A549), and the uptake followed the same trend as-overexpressing HEK293 (FAPα/293) cells. No significant difference was observed between the uptake of ^211^At-FAPI1 and ^211^At-FAPI5 by A549 cells (Figure 3). As expected, there were significant differences in the uptake between the parental and FAPα-overexpressing cell lines for all compounds, from ^211^At-FAPI1 to ^211^At-FAPI5. All FAPI(s) had specificity to FAPα.

### 2.3. Comparison of Body Distributions of FAPI1 or FAPI5 between Two Nuclides

Iodine physiologically accumulates in the thyroid via Na^+^/I^−^ symporter (NIS/SLC7A5), but ^131^I-FAPI1 and ^131^I-FAPI5 accumulated minimally in the thyroid (Table 1). The expression of NIS is also known to be high in the gastrointestinal tract, and high accumulation of both ^131^I-FAPI1 and ^131^I-FAPI5 were observed in the gastrointestinal tract. ^211^At-FAPI1 and ^211^At-FAPI5 also accumulated less in the thyroid, stomach, and kidneys compared with Na^211^At [26], whereas astatine labeled compounds accumulated at higher levels in the thyroid gland 3 h after injection (Table 2). High accumulation of both ^211^At-FAPI1 and ^211^At-FAPI5 was observed in the gastrointestinal tract. The accumulation in tumors of ^211^At-labeled FAPI1 and FAPI5 were more than twice that of ^131^I-labeled FAPI(s). For both nuclides, FAPI compounds were highly excreted in the small intestine and colon. When iodine was used, comparing 1 h (a) and 3 h (b), the amount of excretion increased, but its accumulation in the tumors was low. On the other hand, at 1 h after injection, the urine excretion of ^131^I-FAPI1 was 30.79 ± 15.80% ID, and that of ^131^I-FAPI5 was 9.21 ± 3.52% ID. At 3 h after injection, the urine excretion of ^131^I-FAPI1 was 14.46 ± 4.04% ID, and that of ^131^I-FAPI5 was 3.99 ± 0.91% ID. These values were calculated using the urine retained in the bladder, indicating that ^131^I-FAPI1 was excreted faster. 

At 1 h after ^211^At-FAPI(s) injection, the urine excretion of ^211^At-FAPI1 was 4.92 ± 1.74% ID, and that of ^211^At-FAPI5 was 0.26 ± 0.19% ID. The feces excretion of ^211^At-FAPI1 was 2.68 ± 1.13% ID, and that of ^211^At-FAPI5 was 5.19 ± 1.62% ID. At 3 h after injection, the urine excretion of ^211^At-FAPI1 was 0.03 ± 0.01% ID, and that of ^211^At-FAPI5 was 0.63 ± 0.55% ID. The feces excretion of ^211^At-FAPI1 was 16.44 ± 4.71% ID, and that of ^211^At-FAPI5 was 7.65 ± 2.11% ID. The urine excretion values were calculated using the urine retained in the bladder, which indicated that ^211^At-FAPI1 was excreted faster. After measuring the concentration in each organ, we also measured the rest of the carcass, feces, and floor paper by γ-counter to estimate all distribution of injected dose. While measuring the floor paper, urination was observed, but not much urination was observed after administration. These results showed faster excretion of FAPI1 than FAPI5 and iodine compared with astatine. The difference between FAPI1 and FAPI5 was not marked, and it can be said that they were equivalent. The iodine and astatine levels were clearly different. Particularly in tumors, it was clear that ^211^At-FAPII(s) had superior terms of tumor retention compared with ^131^I-FAPI(s) (Table 1 and Table 2).

### 2.4. Comparison of Anti-Tumor Effects due to Differences in Linker

In terms of therapeutic effects, ^211^At-FAPI1 showed better therapeutic effects. No obvious toxicity was observed for ^211^At-FAPI1 or ^211^At-FAPI5 (Figure 4a). For both compounds, a single administration did not result in significant tumor regression; however, a clear growth inhibitory effect was observed compared to the control (Figure 4b). A significant difference in tumor weight was observed between the groups (control vs. ^211^At-FAPI1, *p* < 0.001; control vs. ^211^At-FAPI5, *p* < 0.05; ^211^At-FAPI1 vs. ^211^At-FAPI5, *p* < 0.05). ^211^At-FAPI1 was found to have a greater tumor growth inhibitory effect than ^211^At-FAPI5 (Figure 4c). At the time of dissection, organs other than the tumor were observed; however, no metastatic tissues were confirmed, and no obvious abnormalities were observed visually. The TLC analysis results of the administered ^211^At-FAPI(s) for the therapeutic experiment are shown in Figure 5. RCPs of 90% or more were ensured in both cases.

## 3. Discussion

During the preparation of ^211^At-FAPI with PEG linker, the radiochemical yields (RCYs) of ^211^At-FAPI1 and 2 were high (100% and 99%, respectively). When producing ^211^At-FAPI5, a byproduct derived from the amino group of PIP was observed (~10%). However, PEG-linked FAPI suppressed the byproduct production. Therefore, non-PIP linkers are suitable for drug manufacturing applications. The RCYs of ^211^At-FAPI3 and 4 were not very high (45% and 15%, respectively), because astatination was carried out at 50 °C to prevent production of a byproduct caused by fluorides (Figure 1a). Additionally, the presence of glucosamine units decreased the astatination reactivity. Based on these results, we presume that ^211^At-FAPI1 is suitable for a stable drug supply.

Cellular uptake and FAPα selectivity were evaluated using an FAPα-overexpressing strain derived from HEK293 cells (parental; HEK293, FAPα overexpression; FAPα/293). We hypothesized that polarity is important for FAPI internalization. Based on ^211^At-FAPI1, we also designed ^211^At-FAPI2, which has a long PEG linker to evaluate difference of FAPα and internalization into cells (Figure 1a). However, FAPα selectivity and cellular uptake did not increase. According to Lindner et al., the presence of fluorine atoms on the pyrrole ring of the FAPI moiety increased its affinity for FAP [4]. Therefore, we compared the FAPα-binding abilities of ^211^At-FAPI1 and 3. Although the RCY of ^211^At-FAPI3 was not suitable, the RCP increased to 94% after purification, which precluded concerns regarding the effects of unlabeled ^211^At components. Unexpectedly, increased FAPα selectivity was not observed (Figure 2). Considering the structure of ^211^At-FAPI13, its hydrophobicity is relatively high compared with that of the FAPI series possessing chelates (e.g., FAPI-02, FAPI-04, FAPI-46, and FAPI-74). Thus, we designed and synthesized ^211^At-FAPI 4 with a highly oxidized glucosamine unit to increase its hydrophilicity. However, the performance did not change (Figure 2).

The results of our investigation indicated that FAPα selectivity and cellular uptake were almost the same for ^211^At-FAPI1 and ^211^At-FAPI5. Considering the molecular size and efficiency of astatination, we employed ^211^At-FAPI1 for comparison with ^211^At-FAPI5 in in vitro experiments. ^211^At-FAPI1 exhibited significant FAPα selectivity, indicating fewer side effects on normal tissues. In contrast, ^211^At-FAPI5 exhibited high cellular uptake, indicating that a high therapeutic effect can be expected at a small dose. Next, we compared the biodistribution of ^211^At-FAPI1 and ^211^At-FAPI5. At 1 h after injection, tumor accumulation of ^211^At- FAPI1 and ^211^At-FAPI5 was 2.15 ± 0.24% ID and 1.40 ± 1.14% ID, which greatly surpassed the accumulation of ^225^Ac-FAPI-04 [22]. After 3 h, the accumulation in tumor cells increased for ^211^At-FAPI1. In contrast, ^211^At-FAPI5 levels decreased (Table 2). Both FAPIs are excreted in the small intestine and urine, and FAPI1 is excreted more rapidly than FAPI5. These trends remained the same even when the nuclides were different. After 1 h, both nuclides and both linkers were present at high doses in the small intestine, and after 3 h, the dose in the colon was high; therefore, they were clearly presumed to be contained in the intestines. Translocation of the ^211^At or ^131^I label to the gastrointestinal contents might be thought to be via digestive juice. Excreted fecal doses were not very high and might have been reabsorbed in the colon. The amount of excreted urine from iodine-labeled FAPIs increased compared with those labeled by astatine; however, its accumulation in tumors was low, and iodine might be unsuitable as a therapeutic agent for labeling FAPI compounds. Compared with FAPI5, FAPI1 exhibited higher retention in tumors and higher non-specific excretion. Because ^211^At could retain more compounds in tumors than ^131^I, we conducted a therapeutic experiment using ^211^At-FAPI1 and 5. Although the data are not shown in this article, it has been confirmed that FAPI uptake into cells is competitively inhibited when an unlabeled FAPI compound is present in excess [24]. Because labeling with ^131^I was inefficient, the possibility that excess unlabeled substances were mixed with the labeled products cannot be ruled out. In other words, the lower % ID/g values for tumors compared to the ^211^At-labeled products in this study may be due to the poor labeling rate. The astatine-labeled PIP linker, the FAPI compound (FAPI5), inevitably produced two spots on the TLC (Figure 5). However, this could not be removed using Oasis HLB column purification (Waters Corporation, Milford, MA, USA). Therefore, it is necessary to investigate the properties of this spot, which is not the main spot. However, this contaminant may explain why tumor accumulation was slightly lower than that of the PEG linker (FAPI1). FAPI5 is difficult to handle because it readily adsorbs onto tubes and syringes. One reason for this is its high structural lipophilicity. This is also an important factor that makes the compound easy to handle for use as a therapeutic agent in nuclear medicine.

Outside Japan, ^225^Ac has attracted attention in alpha-ray nuclear medicine therapy, and clinical trials are underway [27,28,29]. We are also investigating the use of ^225^Ac, but there are still problems with its application in medicine, such as the lack of a mass production method in Japan.

## 4. Materials and Methods

The synthesis procedures for the precursors are described in the Appendix A. We established a synthetic methodology to secure intellectual property. Radiolabeled products were prepared using the following methodology:

### 4.1. Synthesis of ^211^At-FAPI Compounds

^211^At was acquired from the National Institute for Quantum Science and Technology (QST, Takasaki, Gunma, Japan) and RIKEN (Wako, Saitama, Japan) through a supply platform for short-lived radioisotopes. ^211^At was produced according to the ^209^Bi(α, 2n)^211^At reaction and was separated from the Bi target using the dry distillation method. The separated ^211^At was then dissolved in pure water. FAPI(s) with a dihydroxyboryl group (B-FAPI) as a precursor of ^211^At-FAPI compounds were prepared via chemical synthesis according to previously published schemes. ^211^At was produced via a ^209^Bi (α, 2n)^211^At nuclear reaction using a cyclotron. The purification of ^211^At from the irradiated Bi target was performed by dry distillation. All the FAPI compounds were synthesized via a dihydroxyboryl astatine substitution reaction with the corresponding B-FAPI1, 2, 3, 4, and B-FAPI5. In total, 10 µL of aqueous B-FAPI (1 mg/mL), 10 µL of aqueous NaHCO_3_ (7% *w*/*v*), 90 µL of H_2_O, 2–19 µL aqueous ^211^At (0.28–1.01 MBq) and 30 µL of aqueous KI (0.1 mol/L) were added to a polypropylene (PP) tube at room temperature. After being heated to 50 °C or 80 °C for 45 min, the reaction mixture was cooled to ambient temperature. The radiochemical yields (RCYs) of ^211^At-FAPI1, 2, 3, 4, and ^211^At-FAPI5 were 100%, 99%, 45%, 15%, and 90%, respectively. The reaction mixture was then purified using an Oasis HLB column (1.0 mL of H_2_O, 0.5 mL of EtOH/H_2_O (2:3) followed by 0.5 mL of EtOH) to obtain ^211^At-FAPI1-4 and ^211^At-FAPI5. After purification using an HLB column, the radiochemical purities (RCPs) of ^211^At-FAPI1, 2, 3, 4, and ^211^At-FAPI5 were 100%, 100%, 96%, 78%, and 95%, respectively. EtOH was evaporated by heating and replaced with H_2_O for the in vitro and in vivo experiments. We also prepared 5.6–14.1 MBq of ^211^At-FAPI1 and 5.0–17.6 MBq of ^211^At-FAPI5 for the in vivo experiments.

### 4.2. Synthesis of ^131^I-FAPI Compounds

Solutions of [^131^I] NaI were purchased from the Japan Radioisotope Association (JRIA, Kawasaki, Kanagawa, Japan). B-FAPI(s), as precursors of ^131^I-FAPI compounds, were prepared by chemical synthesis (Figure 1b) according to published schemes. All the FAPI compounds were synthesized via a dihydroxyboryl astatine substitution reaction with the corresponding B-FAPI1 and B-FAPI5. Overall, 1 mg of B-FAPI, 80 µL of H_2_O, 28–30 µL aqueous ^131^I-NaI (6.76–6.77 MBq), and 30 µL of 0.4% aqueous *N*-bromosuccinimide (NBS) were added to a polypropylene (PP) tube at room temperature. After heating to 80 °C for 30 min, the reaction mixture was cooled to ambient temperature. The RCY of ^131^I-FAPI1 and ^131^I -FAPI5 are 13% and 51%. After adding 4% aqueous ascorbic acid (10 µL), the mixture was stirred for 15 min. Then, the reaction mixture was purified using an Oasis HLB column (1.0 mL of H_2_O, 0.5 mL of EtOH/H_2_O (2:3) followed by 0.5 mL of EtOH) to obtain ^131^I-FAPI1 and ^131^I-FAPI5. After purification using an HLB column, the radiochemical purities (RCPs) of ^131^I-FAPI1 and ^131^I-FAPI5 were 91% and 99%, respectively. EtOH was evaporated by heating and replaced with H_2_O for the in vitro and in vivo experiments. 

### 4.3. In Vitro Cellular Uptake Analysis

HEK293 (human embryonic kidney) and A549 (human lung cancer) cell lines were provided by the RIKEN Cell Bank. PANC-1 cells (human pancreatic cancer) were obtained from the American Type Culture Collection (ATCC). The cells were maintained in Dulbecco’s modified Eagle’s medium: D-MEM (FUJIFILM Wako Pure Chemical Corporation, Osaka, Japan) supplemented 10% heat-inactivated fetal bovine serum (FBS, Gibco, Thermo Fisher Scientific, Waltham, MA, USA) and 1% penicillin–streptomycin (FUJIFILM Wako Pure Chemical) or Eagle’s minimal essential medium: E-MEM) supplemented with 10% FBS, 1% antibiotics, and 1% non-essential amino acids (NEAA, FUJIFILM Wako Pure Chemical). A lentiviral vector plasmid containing the human FAPα gene clone was purchased from Applied Biological Materials, Inc. (Richmond, BC, Canada). This plasmid is based on pLenti-GIII-CMV-RFP-2A-Puro and contains a red fluorescent protein (RFP) tag. After infection with the vector, puromycin-resistant clones of all cells (HEK293/FAPα and A549/FAPα) as the experimental group were obtained. We confirmed the expression of FAPα and RFP by flow cytometry (Attune NxT, Thermo Fisher Scientific) (Appendix A). Parental cells and FAPα overexpressing cells were seeded into 24-well plates (5 × 10^4^/well) and cultured for two days. After two washes with phosphate-buffered saline without calcium and magnesium (PBS (−)), the culture medium was replaced with HBSS (+). After treatment with ^211^At-labeled FAPI series compounds, cells were washed twice with PBS (−). After washing, all cells were lysed with NaOH (0.1 N) and the radioactivity of the cells was counted using a 2480 Wizard^2^ γ counter (Perkin Elmer, Waltham, MA, USA). Protein levels were measured using a plate reader (MultiScan FC; Thermo Fisher Scientific) and a BCA protein assay kit (FUJIFILM Wako Pure Chemical). Uptakes (CPM/µg protein) were compared between parental cells and cells overexpressing FAPα at 30 min after the incubation with ^211^At-labeled FAPI series compounds.

### 4.4. Preparation of Animals

The animals were prepared according to previously published protocol [30]. The PANC-1 cell line was obtained from the ATCC. Male BALB/c-nu-nu (nude) mice were purchased from Japan SLC Inc. (Hamamatsu, Japan). PANC-1 cells were cultured in a DMEM containing 10% fetal bovine serum and 1% penicillin–streptomycin at 37 °C in a humidified incubator with 5% CO_2_. The cultured cells were washed with phosphate-buffered saline (PBS) and harvested using trypsin. Tumor xenograft models were established by subcutaneous injection of 1 × 10^7^ cells suspended in 0.2 mL culture medium and Matrigel^®^ (1:1; BD Biosciences, Franklin Lakes, NJ, USA) into nude mice. Animals with tumors >50 mm^3^ in size were used. This study was approved by the Osaka University Graduate School of Science Animal Care and Use Committee (permission number: 2020-02-0) and was conducted according to the Osaka University Animal Experimentation Regulations.

### 4.5. Measurement of Biodistribution of FAP1 and FAPI5 Labeled with ^131^I or ^211^At

The compounds of interest were labeled with ^211^At or ^131^I. Mice were dissected 1 or 3 h after injection. The radioactivities were measured with a γ counter (Wizard^2^ 2480) and the organ weights were also measured. Using these data, the % ID/g was calculated.

#### 4.5.1. Biodistribution of ^131^I-FAPI1 and ^131^I-FAPI5

PANC-1 xenograft mice were divided into groups according to the tumor size. The average body weight of the ^131^I-FAPI1 group was 22.07 ± 0.19 g, and that of the ^131^I-FAPI5 group was 22.25 ± 0.18 g. The mice were treated with the following conditions:^131^I-FAPI1 (0.12 ± 0.001 MBq), ^131^I-FAPI5 (0.27 ± 0.01 MBq). One or three hours after injection, the mice were euthanized and dissected, radioactivity was measured with a counter, and the weights of the organs were measured using a microbalance. 

#### 4.5.2. Biodistribution of ^211^At-FAPI1 and ^211^At-FAPI5

Animals were prepared according to previously described protocols. The average body weight of the ^211^At-FAPI1 group was 22.25 ± 0.18 g, and that of the ^211^At-FAPI5 group was 22.07 ± 0.19 g. The animals were treated with the following conditions:^211^At-FAPI1 (0.81 ± 0.02 MBq) and ^211^At-FAPI5 (0.54 ± 0.03 MBq). After the injection, the experimental mice were treated using the same procedure as for ^131^I.

### 4.6. Examination of Therapeutic Effect

The animals were prepared according to previously published protocol [30]. PANC1 itself did not express a high level of FAPα, but it had already been confirmed that FAPα was highly expressed in xenograft tumors [22]. PANC-1 xenograft model is commonly used to evaluate the therapeutic efficacy of FAPIs. This model represents FAPα at a low level in the cancer cells themselves, but on the other hand, it well represents the state in which FAPα is highly expressed in the cancer tissue. Mice were divided as follows: control (20.88 ± 0.50 g: saline was used for control), ^211^At-FAPI5 (20.95 ± 0.62 g:0.96 ± 0.06 MBq), ^211^At-FAPI1 (22.25 ± 0.31 g:0.97 ± 0.08 MBq). The mice were monitored three times per week, and their body weights and tumor sizes were measured. The animals were euthanized based on humane endpoints, but can be euthanized even when they are in good health, when tumors exceed 10% of body weight, or when tumors become necrotic. Based on these criteria, the observation period was four weeks.

### 4.7. Statistical Analysis

The results are expressed as the means ± the standard error. Comparisons between groups were performed using the unpaired *t*-test and Mann–Whitney U test in Microsoft Excel for Mac (ver. 16.71). For multiple comparisons among the three groups, Bonferroni correction was performed. Differences were considered statistically significant at *p* < 0.05.

## 5. Conclusions

We evaluated the cellular uptake and FAPα selectivity of ^211^At-FAPI1 to 4. No differences in uptake according to PEG length were observed. Based on the results of cellular uptake, nuclide labeling efficiency, and in vivo pharmacokinetics, simple PEG-linker compounds (FAPI1) exhibited the best properties. In the future, we plan to use this compound to screen more effective treatment methods and applicable cancer types. 

## Figures and Tables

**Figure 1 ijms-24-08701-f001:**
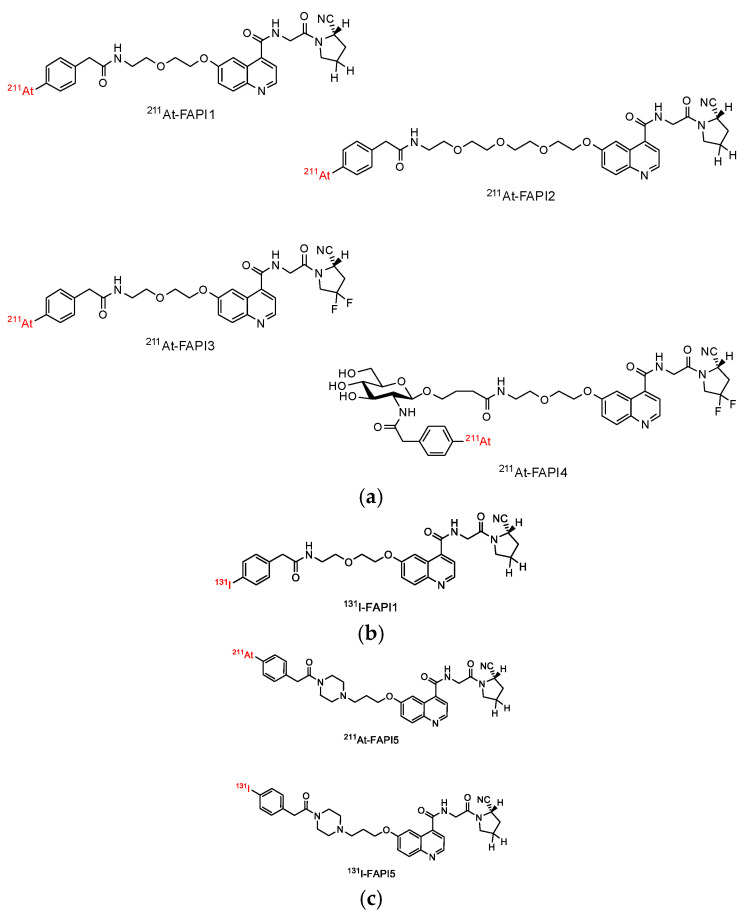
Structure of ^211^At-FAPI and ^131^I-FAPI compounds. (**a**) ^211^At-labeled PEG linker FAPI(s) were numbered from 1 to 4 according to their structural complexity. (**b**) ^131^I-labeled PEG linker compound. (**c**) PIP linker compound. PIP linker compound was synthesized based on the already reported compound for comparison with the PEG linker compound.

**Figure 2 ijms-24-08701-f002:**
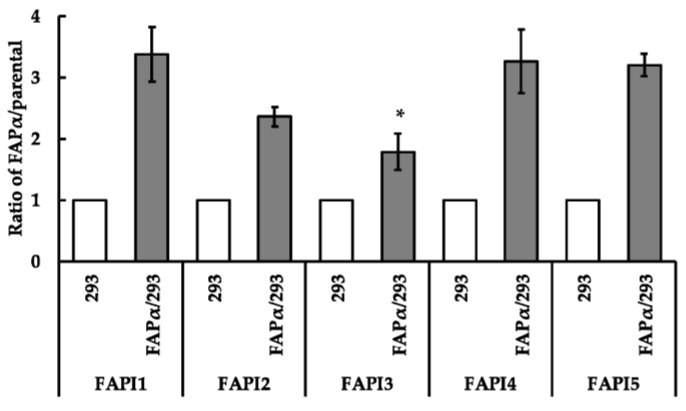
Cellular uptake of ^211^At-FAPI1-5. The y-axis represents the ratio divided intensity of the FAPα overexpressing cell line with those of the parental cell line. HEK293 cells were used for this experiment, with 293 indicating HEK293 parental and FAPα/293 indicating FAPα-overexpressing HEK293 cell line. This histogram represents means ± S.E. An asterisk attached to the histogram indicates a significantly different ratio of FAPI1 in FAPα/293. * *p* < 0.05.

**Figure 3 ijms-24-08701-f003:**
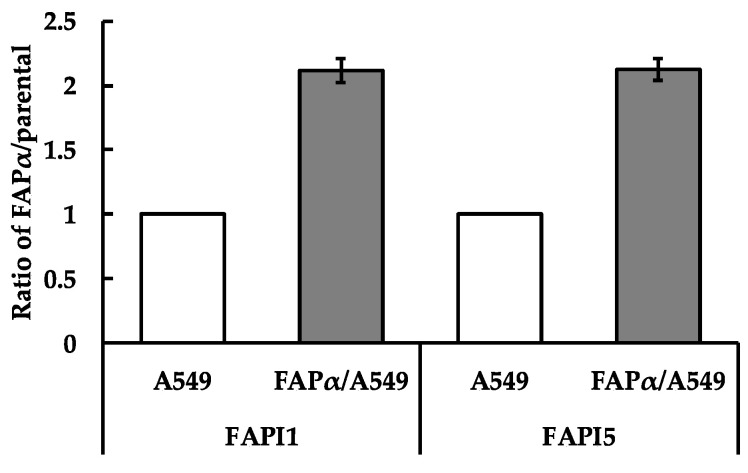
Cellular uptake of ^211^At-FAPI1 and 5. The y-axis represents the ratio divided intensity of the FAPα-overexpressing cell line with those of the parental cell line. A549 cells were used for this experiment, with A549 indicating A549 parental and FAPα/A549 indicating the FAPα-overexpressing A549 cell line. This histogram represents means ± S.E.

**Figure 4 ijms-24-08701-f004:**
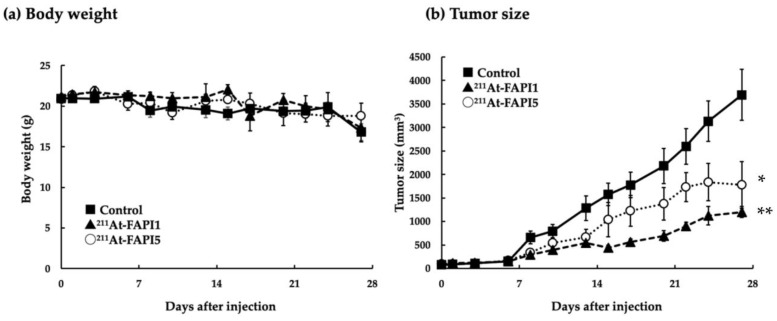
Anticancer effect in PANC-1 xenograft mice after administration of ^211^At-labeled FAPI (approximately 1 MBq). (**a**) Body weight of mice. (**b**) Tumor sizes of the experimental mice. Means ± S.E. Filled squares are the control group, filled triangles are the ^211^At-FAPI1 group, and white circles are the ^211^At-FAPI5 group. * *p* < 0.05, ** *p* < 0.01, *** *p* < 0.001, and # *p* < 0.05. (**c**) Tumor weights of each group. Each group consisted of four mice.

**Figure 5 ijms-24-08701-f005:**
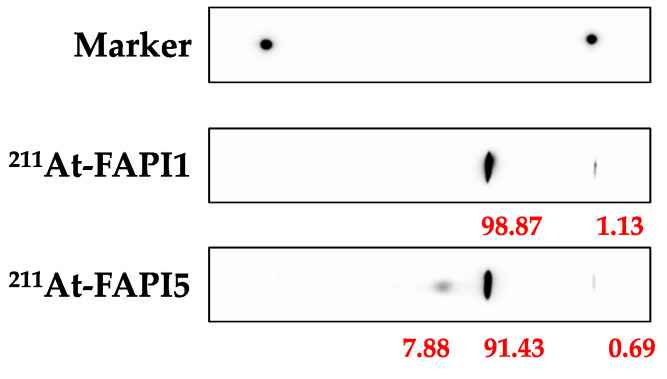
The quality evaluation data of ^211^At-FAPI1 and 5. The eluent result of TLC is shown. ^211^At-FAPI1, 5, and control solutions were spotted on the silica membrane. The membranes were treated in water: ACN = 1:2 solution. TLC: thin-layer chromatography, ACN: Acetonitrile.

**Table 1 ijms-24-08701-t001:** Distribution of ^131^I-FAPI1 and ^131^I-FAPI5 at (**a**) 1 h and (**b**) 3 h after injection. Carcass was the sample that measured all of the remainder of the dissected mouse. % ID/g was calculated. All groups consisted of three mice. (% ID: % of injection dose) * *p* < 0.05.

(a) 1 h			(b) 3 h		
	^131^I-FAPI1	^131^I-FAPI5		^131^I-FAPI1	^131^I-FAPI5
Brain	0.54 ± 0.15	0.50 ± 0.10	Brain	0.34 ± 0.04	0.30 ± 0.02
Thyroid	7.96 ± 1.99	5.66 ± 0.26	Thyroid	12.86 ± 0.68 *	8.40 ± 0.70
Salivary gland	2.32 ± 0.12	2.65 ± 0.36	Salivary gland	1.72 ± 0.30	1.82 ± 0.11
Heart	1.33 ± 0.06	1.21 ± 0.07	Heart	0.91 ± 0.12	1.95 ± 0.06
Lung	3.82 ± 2.11	4.43 ± 2.34	Lung	2.15 ± 0.47	2.33 ± 0.38
Liver	7.02 ± 0.90	7.40 ± 0.71	Liver	3.02 ± 0.70	2.85 ± 0.29
Stomach	7.67 ± 3.50	7.75 ± 4.99	Stomach	2.57 ± 0.66	2.10 ± 0.84
Small Intestine	48.18 ± 23.44	38.53 ± 17.18	Small Intestine	5.99 ± 1.91	5.54 ± 1.90
Colon	9.63 ± 1.69	10.43 ± 1.67	Colon	155.39 ± 16.65	165.96 ± 15.26
Kidney	3.68 ± 0.26	3.92 ± 0.41	Kidney	1.74 ± 0.12	1.74 ± 0.08
Pancreas	1.54 ± 0.18	1.58 ± 0.30	Pancreas	0.63 ± 0.12	0.79 ± 0.12
Spleen	1.42 ± 0.42	1.15 ± 0.06	Spleen	0.54 ± 0.17	0.64 ± 0.05
Testis	0.56 ± 0.04	0.66 ± 0.08	Testis	0.76 ± 0.13	0.64 ± 0.08
Blood	1.99 ± 0.20	2.36 ± 0.17	Blood	2.86 ± 0.46	2.32 ± 0.19
Bone	0.47 ± 0.07	0.68 ± 0.09	Bone	0.37 ± 0.07	0.40 ± 0.01
Tumor	0.77 ± 0.31	1.13 ± 0.10	Tumor	0.89 ± 0.44	1.13 ± 0.07

**Table 2 ijms-24-08701-t002:** Distribution of ^211^At-FAPI1 and ^211^At-FAPI5 at (**a**) 1 h and (**b**) 3 h after injection. After the mice were dissected, the intensities were measured and the % ID/g were calculated. All data are shown as means ± SE. All groups consisted of three mice. (% ID: % of injection dose.) Significant differences by Mann–Whitney U test are shown as follows. * *p* < 0.05.

(a) 1 h			(b) 3 h		
	^211^At-FAPI1	^211^At-FAPI5		^211^At-FAPI1	^211^At-FAPI5
Brain	0.38 ± 0.04	0.30 ± 0.05	Brain	0.20 ± 0.04	0.30 ± 0.07
Thyroid	3.73 ± 0.57	3.58 ± 0.71	Thyroid	16.12 ± 5.84	24.33 ± 6.86
Salivary gland	4.18 ± 0.72	3.27 ± 0.38	Salivary gland	8.51 ± 2.25	6.43 ± 3.55
Heart	3.24 ± 0.17 *	1.51 ± 0.14	Heart	2.21 ± 0.13	0.90 ± 0.63
Lung	4.61 ± 0.68	2.86 ± 0.24	Lung	4.92 ± 1.39	3.28 ± 1.95
Liver	17.06 ± 0.98 *	6.46 ± 0.62	Liver	1.77 ± 0.44	0.50 ± 0.61
Stomach	19.31 ± 1.21	14.33 ± 3.00	Stomach	9.30 ± 1.27 *	1.85 ± 1.70
Small Intestine	33.16 ± 3.61	28.20 ± 5.27	Small Intestine	4.85 ± 1.28	4.89 ± 3.82
Colon	6.34 ± 0.35	4.29 ± 0.44	Colon	32.44 ± 14.99	58.40 ± 9.91
Kidney	6.92 ± 1.34 *	3.07 ± 0.07	Kidney	2.22 ± 0.24	1.56 ± 0.81
Pancreas	1.43 ± 0.16	1.17 ± 0.12	Pancreas	0.63 ± 0.05	0.45 ± 0.29
Spleen	3.93 ± 0.52	2.58 ± 0.51	Spleen	5.27 ± 1.11	4.32 ± 0.21
Testis	1.19 ± 0.10	1.05 ± 0.07	Testis	1.51 ± 0.28	1.95 ± 0.46
Blood	3.76 ± 2.69	1.32 ± 0.04	Blood	1.20 ± 0.11	1.33 ± 0.24
Bone	1.07 ± 0.12	0.61 ± 0.13	Bone	0.70 ± 0.14	1.35 ± 0.51
Tumor	2.15 ± 0.24 *	1.24 ± 0.14	Tumor	3.04 ± 0.69 *	1.48 ± 0.89

## Data Availability

Data are contained within the article and the Appendix A.

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
