# Peer review of "Evaluation of Astatine-211-Labeled Fibroblast Activation Protein Inhibitor (FAPI): Comparison of Different Linkers with Polyethylene Glycol and Piperazine"

_ijms, 2023, doi:10.3390/ijms24108701_

Round 1

Reviewer 1 Report

The manuscript entitled “Evaluation of astatine-211-labeled fibroblast activation protein inhibitor (FAPI): comparison of different linkers with polyethylene glycol and piperazine” describes the preparation and assessment of FAPI derivatives aiming to achieve improved stability for the development of efficient FAPI-targeted therapy.

The existent FAPI tracers (i.e. FAPI74 and 46 and close derivatives) have evident intrinsic structural flaws and more stable alternatives are definitely in demand.

The manuscript requires the following multiple major amendments and additions to be considered for publication:

Major:

A) Multiple crucial subjects in the text are barely developed (especially in the Introduction and Discussion sections). In general, the manuscript is badly structured and would significantly benefit from a serious rearranging, and re-elaboration.

The Introduction section has an abrupt beginning (a proper description of the target and the type of molecules of interest is totally missing) and has a very patchy structure. A clear rationale behind the work is missing. The authors should add more details and elaborate further the various basic concepts: it is good to be concise but tackling an entire subject in just one short sentence is an impossible task and it is not helping the reader understanding the message properly (for example, and not exclusively, Page 2, line 71 “Thus FAPI is a promising molecule for cancer treatment”, or line 86: “This astatination method did not require toxic reagents”, or Page 6, line 172: “211At-FAPI with PEG linker”).

B)  The authors need to add the following important information and amend the following imprecisions:

1)    Page 2, line 89: The authors state that “piperazine (the chemical?) is degraded in the body”. Do the authors have any evidence of that or do they mean that the piperazine-containing-FAPI (and not just “piperazine”) is unstable in vivo due to the presence of the piperazine linker? The authors should either produce the suitable references or correct the sentence accordingly (and avoid any misinformation and confusion).

2)    Page 2, Lines 80-82: Unnecessary self-citation: the authors add a long list of references (reference 23-27) which are not crucial for the story and can be omitted. To prove the author’s point, reference 29 should be enough since it can perfectly show the authors’ experience with both the use of 211-At and FAPIs.

3)    The Results section is not pleasant to read. It is patchy and needs a more detailed and coherent (i.e., structured) narrative.

No results about the production of the radiolabeled products (one of the main subjects of the manuscript) have been described. The authors must add few lines about it (there is just a short mention of RCYs in the Material and Methods section, but it is insufficient and in the wrong place).

The biodistribution data from tables in Section 2.2 (page 3) have been barely mentioned and described. Please add few lines.

Section 2.3: “No obvious toxicity was observed”: the authors should elaborate this concept further.

How was the 131- and 211-At-FAPI dose (given to the mice) decided? Was it based on previous data (in this case, a reference should be given)? Or after performing a dose escalation? The authors should spend few words on the subject.

4)    Page 4, lines 128-132: The authors estimate the urine excretion by measuring the activity retained in the bladder after 1h and 3h p.i. (only). This is not the standard (and most appropriate) method to determine the renal excretion since it does not take in consideration the urine expelled by the mice by urination during the 1-3 h interval. Why do the authors performed the test in this way (a method that could give just a very rough estimation of the urine excretion) instead of a proper study using metabolic cages? The authors should disclose that the results achieved in this way are just a rough estimation and should indicate why just a rough estimation was enough for them.

5)    Page 5, line 159: After treatment, the authors removed the tumors and weighted them. Did the authors perform any IHC on the tumors? If yes, please add the staining figures and a proper description. If not, why not? IHC gives very insightful and important information, especially after therapy.

6)    Page 5, Figure 5, page 7, lines 220-221, Figure S1: TLC analysis is a quite crude technique and could be sufficient for the analysis of radiolabeled big molecules (such as proteins). However, for small molecules (such as FAPIs) basic TLC evaluation is usually complemented by RP-HPLC analysis which could provide a more detailed picture of the (radio)chemical purity and integrity of the radiolabeled product. Is there a reasonable motivation for the lack of HPLC analysis in the manuscript? If yes, the authors should add few words about it in the text.

7)    Discussion section: The first paragraph is patchy, disconnected and has no clear flow. The authors should re-structure this part.

8)    Page 6, lines 199-201: “211At-FAPI1 exhibited ….”. The authors state some conclusions (i.e. lower side effects on normal tissues and high therapeutic effects) which are difficult to draw from the very small number and array of experiments that have been described in the manuscript. Please, either remove the sentence or add additional experimental evidence to corroborate the statement.

9)    Page 7, line 212: The sentence “Because 211At could retain more compound in tumors than 131I” has no sense. How can a radioisotope retain a compound inside the tumor? Please correct the sentence.

10) Page 7, lines 220-221 and Tables S1and S2: The RCPs of both 211At-FAPI4 (78%) and 131I-FAPI1 (73%) are suboptimal (as also acknowledged by the authors themselves) and clearly demonstrate that the purification method used by the authors is not fit for purpose for those two compounds. Did the authors try any other purification methods apart from HLB-SPE? Any other type of SPE cartridges? What about purification by HPLC?

Also, products with such suboptimal purity should not have been used in in vitro/vivo tests and should have not been reported.

11) Page 9, line 293: The authors state that they confirm the FAP expression status by flow cytometry. Please show the data (e.g., histograms) in Supporting Info.

12) Page 9, section 4.4: An ethic statement for the work using animals should be added (see MDPI “Instructions for Authors”).

13) Supporting Information: Please supply the NMR and MS spectra.

Why Imidate S8 (page 7) has no 13C chemical shifts? Also, “colourless” is a more suitable adjective than “white” (line 179).

A general list of abbreviations should be added since the authors use multiple abbreviations without defining them (e.g. NMP, HOAt, AcOH, CCl3CN, MS4A…).

When authors indicate the yield as “quantitatively” (page 5, line 143; page 7, line 204), the amount of product (in mg) is missing. Please add.

Page 9 and 10, Tables S1 and S2: The values of RCY and RCP have no range or SD values giving the impression that the products were obtained only once. Was the reaction done only once (n = 1) or more times? If more than one radiolabeling was performed, please indicate the range of RCYs and RCPs values.

Please add the type (volume and amount of sorbent) and the supplier of the Oasis HLB columns.

C) Minor:

Page 2, line 92: Why Figure 1 is in page 8 after all the other figures? Please either move Figure 1 before Figure 2 or change the numbering accordingly.

Page 3, Line 105 (and other places throughout the manuscript): Is FAPII the same as FAPα?

Page 7, Material and Method: The radiolabeling procedure is described in the main manuscript and there is no reason why the same method is reported also in Supporting information.

Page 7, lines 234 and 337: the cyclotron production of 211-At is repeated twice.

Page 8, line 257: Please add the reference for the published schemes.

Page 9, lines 282-283: PANC-1 cells were used for the in vivo work and not for the in vitro binding assessment. The sentence should be moved to the proper sub-section.

Page 10, Line 341: Please indicate how the tumor sizes were measured.

Page 12: References 18 and 19: the titles of the referenced papers are incomplete.

In general, the English is readable, the structure of the sentences should be improved.

Reviewer 2 Report

The paper under investigation is an interesting original paper that investigates some novel 211At-FAPI(s) possessing a polyethylene glycol (PEG) linker. The manuscript is focused on a relevant topic.

Some considerations:

Abstract is very short and does not clearly reflect the findings reported in the paper.

Please be sure to put every acronym also in the extenso form: e.g. page 2, line 50 EC50

Some sentences in the introduction are very short, making this section a little too much  syncopated and difficult to be read, e.g. Biodistribution of 68Ga-DOTA.SA.FAPI in patients with cancer was also demonstrated, and clear accumulation was shown by PET/CT imaging [18]. [68Ga] Ga-66 DATA5m.SA.FAPI is a promising substrate for PET/CT imaging [19]. Etc.. Please revise.

Page 2, the authors state: “Therefore, hospitalization is not required, and this contributes to the improvement of the quality of life of cancer patients.” I find this sentence misleading. Hospitalization might depend on different issues, aside from the type of employed radionuclide, such as therapy tolerability from patients. Since the majority of studies on targeted alpha therapy are at an embryonic phase, it is too early to state that hospitalization is not required. Please amend.

Discussion

In general, I would suggest to improve writing quality to make the manuscript more readable. In addition, I would stress the translational implications of the manuscript, particularly concerning the usefulness of targeted alpha therapy with 211At with respect to 225Ac-labeled compounds, already under investigations in some clinical trials (PMID: 34175980, PMID: 36864360, PMID: 32820953), as briefly mentioned by the same authors in the Introduction.

Round 2

Reviewer 2 Report

The authors have properly addressed reviewers' comments.